# The Elevated Inflammatory Status of Neutrophils Is Related to In-Hospital Complications in Patients with Acute Coronary Syndrome and Has Important Prognosis Value for Diabetic Patients

**DOI:** 10.3390/ijms25105107

**Published:** 2024-05-08

**Authors:** Elena Barbu, Andreea Cristina Mihaila, Ana-Maria Gan, Letitia Ciortan, Razvan Daniel Macarie, Monica Madalina Tucureanu, Alexandru Filippi, Andra Ioana Stoenescu, Stefanita Victoria Petrea, Maya Simionescu, Serban Mihai Balanescu, Elena Butoi

**Affiliations:** 1Department of Cardiology, Elias Emergency Hospital, 011461, Carol Davila University of Medicine and Pharmacy, 050474 Bucharest, Romania; andra.copaciu@gmail.com (A.I.S.); smbala99@gmail.com (S.M.B.); 2Biopathology and Therapy of Inflammation, Institute of Cellular Biology and Pathology “Nicolae Simionescu”, 050568 Bucharest, Romania; andreea.mihaila@icbp.ro (A.C.M.); anca.gan@icbp.ro (A.-M.G.); letitia.ciortan@icbp.ro (L.C.); razvan.macarie@icbp.ro (R.D.M.); monica.pirvulescu@icbp.ro (M.M.T.); maya.simionescu@icbp.ro (M.S.); 3Department of Biochemistry and Biophysics, “Carol Davila” University of Medicine and Pharmacy, 020021 Bucharest, Romania; alexandru.filippi@gmail.com; 4Nephrology Hospital Dr. Carol Davila, 010731 Bucharest, Romania; p_stefanita@yahoo.com

**Keywords:** neutrophils, NETs, acute coronary syndrome, diabetes, prognostic score

## Abstract

Despite neutrophil involvement in inflammation and tissue repair, little is understood about their inflammatory status in acute coronary syndrome (ACS) patients with poor outcomes. Hence, we investigated the potential correlation between neutrophil inflammatory markers and the prognosis of ACS patients with/without diabetes and explored whether neutrophils demonstrate a unique inflammatory phenotype in patients experiencing an adverse in-hospital outcome. The study enrolled 229 ACS patients with or without diabetes. Poor evolution was defined as either death, left ventricular ejection fraction (LVEF) <40%, Killip Class 3/4, ventricular arrhythmias, or mechanical complications. Univariate and multivariate analyses were employed to identify clinical and paraclinical factors associated with in-hospital outcomes. Neutrophils isolated from fresh blood were investigated using qPCR, Western blot, enzymatic assay, and immunofluorescence. Poor evolution post-myocardial infarction (MI) was associated with increased number, activity, and inflammatory status of neutrophils, as indicated by significant increase of Erythrocyte Sedimentation Rate (ESR), C-reactive protein (CRP), fibrinogen, interleukin-1β (IL-1β), and, interleukin-6 (IL-6). Among the patients with complicated evolution, neutrophil activity had an important prognosis value for diabetics. Neutrophils from patients with unfavorable evolution revealed a pro-inflammatory phenotype with increased expression of *CCL3*, *IL-1β*, *interleukin-18 (IL-18)*, *S100A9*, *intracellular cell adhesion molecule-1 (ICAM-1)*, *matrix metalloprotease (MMP-9)*, of molecules essential in reactive oxygen species (ROS) production *p22phox* and *Nox2*, and increased capacity to form neutrophil extracellular traps. Inflammation is associated with adverse short-term prognosis in acute ACS, and inflammatory biomarkers exhibit greater specificity in predicting short-term outcomes in diabetics. Moreover, neutrophils from patients with unfavorable evolution exhibit distinct inflammatory patterns, suggesting that alterations in the innate immune response in this subgroup may exert detrimental effects on disease progression.

## 1. Introduction

The role of chronic inflammation in cardiovascular and metabolic diseases is nowadays well established by extensive preclinical and clinical research. Trials such as CANTOS, COLCOT, and LoDoCo demonstrated that anti-inflammatory therapies in cardiovascular disease (CVD) reduce the occurrence of a non-fatal myocardial infarction (MI), non-fatal stroke, or cardiovascular death [1,2,3]. Although interest in the role of inflammation in patients with coronary atherosclerosis now extends from a chronic perspective to an acute approach, research in this matter is less advanced. From a simplistic perspective, a clinician can easily observe the inflammatory response in patients with acute MI using only the usual blood tests, such as neutrophils and C-reactive protein (CRP). However, the complexity of the immune response in the acute phase makes a practical diagnostic, prognostic, or therapeutic approach difficult now. It is still challenging to establish the point from which the acute cytokine cascade stops being a useful repairing stimulus and becomes a harmful mechanism.

Several risk scores are currently used for in-hospital mortality in patients with MI, such as GRACE (Global Registry of Acute Cardiac Events) and TIMI (Thrombolysis in Myocardial Infarction), but they do not include inflammation parameters [4,5]. In addition, the current prediction scores do not distinguish between diabetic and non-diabetic patients, although different parameters seem to influence the evolution. The poorer outcomes in diabetic patients can be attributed to an elevated pro-inflammatory and prothrombotic state [6].

Although the in-hospital mortality rate has diminished since the introduction of standardized early revascularization for all patients with ST-elevation myocardial infarction (STEMI), there is still heterogeneity in response to ischemia and reperfusion, and consequently also in the evolution of patients with acute MI. Even though the role of inflammation and its close relationship with ischemia-reperfusion injury and adverse cardiac remodeling in MI has been extensively explored [7], gaps in the knowledge of the complex interactions between these processes still exist.

Accumulating clinical evidence underscores the potential involvement of innate immune cells in the development of acute coronary syndrome (ACS), and emerging findings indicate a simultaneous alteration in the phenotype of neutrophils during ACS [8]. Moreover, an increased neutrophil-to-lymphocyte ratio (NLR) has been linked to CVD and its near-term adverse outcomes. In acute, dynamic changes in NLR precede the clinical state by several hours and can provide rapid data on prognosis.

Following MI, neutrophils are quickly recruited to the ischemic region, where they initiate the inflammatory response, cleaning up dead cell debris. In general, these cells are crucial for defending the host against threats using different weapons: antimicrobial enzymes, oxidative burst, and the release of neutrophil extracellular traps (NETs). Nonetheless, the very mechanisms that empower them to eliminate pathogens and enhance inflammatory responses can also lead to tissue damage. A pertinent scenario emerges following MI, wherein an intricate interplay occurs. Depending on the delicate equilibrium or undue immune response, neutrophils can serve a dual purpose: they might initiate vital reparative pathways essential for scar formation, or they might inadvertently induce damage that subsequently sets off a cascade of maladaptive remodeling processes [9].

Recruited neutrophils orchestrate the inflammatory process through the production and release of a diverse array of signaling molecules, such as cytokines/chemokines, reactive oxygen species (ROS), and proteolytic enzymes that cause tissue injury [10]. In addition, activated neutrophils form NETs and release extracellular vesicles that contain a multitude of inflammatory mediators. NETs promote occlusion of the vasculature and thrombosis and have been shown to be cytotoxic, pro-inflammatory and pro-thrombotic, potentially augmenting myocardial injury [11]. Recently, it has been suggested that NETs influence both the size of myocardial infarctions and the subsequent remodeling of the left ventricle following a heart attack [12].

Although the researchers reported that neutrophils are closely related to the severity and prognosis of patients with MI, and NLR in post-MI patients had predictive value for major adverse cardiac events, if an altered neutrophil phenotype in these patients is responsible for negative prognosis, it is not known.

In this paper, we investigated the involvement of inflammatory parameters of MI patients during the index event hospitalization and evaluated the pro-inflammatory phenotype of neutrophils isolated from MI patients with or without negative evolution. Elucidating if a specific neutrophil phenotype defined by specific pro-inflammatory markers is responsible for the negative prognosis of MI patients is highly relevant for further development of potential anti-inflammatory treatment.

## 2. Results

### 2.1. Patient Groups and Demographics

Clinical data for the whole study population, as well as for the control, pre-diabetes and diabetes subgroups of patients, are summarized in Table 1. The subgroups had similar sex distributions, with about one-third female and two-thirds male participants and no significant differences in age (*p* > 0.05 for all intergroup comparisons, ANOVA, Tukey’s post hoc test). Diabetic patients had a higher percentage of high blood pressure compared with both pre-diabetic patients (chi-square, *p* = 0.0173) and non-diabetic patients (*p* = 0.0067), as well as a higher prevalence of prior MI (chi-square, *p* = 0.0122 and *p* = 0.0015 vs. pre-diabetic and non-diabetic patients, respectively). In addition, the patients with diabetes had significantly lower low-density lipoprotein (LDL) values compared with non-diabetic patients (*p* = 0.0493, one-way ANOVA, Tukey’s post hoc test), likely due to stricter drug control of LDL levels in these patients. No other significant differences were found between groups.

### 2.2. Clinical and Paraclinical Factors Associated with Poor Prognosis

A multinomial logistic regression conducted on clinical data to evaluate factors predicting poor prognosis revealed that anemia (HR = 3.1, *p* = 0.045), atrial fibrillation (HR = 3.5, *p* = 0.013), and diabetes (HR = 2, *p* = 0.032) are independent risk factors significantly influencing the early outcome of MI (see Figure 1A).

In our quest to identify biomarkers linked to unfavorable prognosis, we assessed 32 paraclinical parameters, including both biochemical and cytology markers. The full list of these parameters is presented in Table 2. We compared the levels of these parameters between patients with favorable and unfavorable outcomes. The results showed that patients within the unfavorable evolution group had higher values for parameters associated with: (i) clot formation: D-Dimers (*p* = 0.0119, Figure 1D), fibrinogen (*p* = 0.0035, Figure 1C); (ii) inflammation: erythrocyte sedimentation rate (ESR) (*p* = 0.0027), CRP (*p* = 0.0021, Figure 1B), and (iii) renal dysfunction: creatinine (*p* = 0.0024, Figure 1E). The interleukin-6 (IL-6) and interleukin-1β (IL-1β) were also modified in this group (Appendix A). Moreover, in patients with unfavorable evolution, lymphocytes were reduced in number, neutrophils number increased (Figure 1F), and so, neutrophils to lymphocyte ratio, NLR, was also highly elevated (*p* < 0.001, Figure 1G). Notably, the neutrophil products and markers of activation, NETs (*p* = 0.0297, Figure 1H) and S100A8/9 (*p* = 0.0293, Figure 1I) were increased in patients with unfavorable in-hospital evolution—the data suggesting that neutrophil activation has a role in determining the patient outcome.

Neutrophils (along with derivative white blood count [WBC] and NLR parameters), NETs, S100A8/9, and Troponin formed a cluster of highly inter-correlated parameters (Figure 2A). In this cluster, troponin, a strong independent predictor in myocardial infarction, was positively correlated only with neutrophil or neutrophil-associated inflammatory molecules (NLR, S100A8/A9, NETS and IL-6) but not with ESR, Ferritin, Fibrinogen, and CRP (Figure 2A). As expected, maximum quantitative troponin levels, as a marker of the volume of myocardial necrosis, were negatively correlated with left ventricular ejection fraction (LVEF) (Figure 2A). Furthermore, we obtained a strong negative correlation between neutrophils or NLR and admission LVEF (Figure 2A).

While NLR, D-Dimers, NETs, and S100A8/9 were associated with an unfavorable evolution, as shown above, the diabetic status was not associated with significantly modified levels of either of these parameters (Figure 2D–G).

### 2.3. Prognosis Scores

Given that diabetic patients showed a higher probability of experiencing an unfavorable evolution (Figure 1A) without demonstrating significantly elevated levels of parameters associated with poor outcomes in the overall study population (NLR, D-Dimers, NETs, S100A8/A9), we formulated the hypothesis that these paraclinical parameters may have different clinical implications for patients with and without diabetes. Therefore, we employed linear Logistic Regression to calculate the sensitivity and specificity of all 32 parameters in predicting unfavorable prognoses for non-diabetic and diabetic patients. Based on the sensitivity and specificity calculated at each parameter value, a Receiver Operating Characteristic (ROC) curve analysis was performed to identify threshold values as the values that best discriminate between favorable and unfavorable prognoses. Thus, for each parameter, the Youden j index (Sensitivity + Specificity −1) was calculated for each possible value that the parameter took, and the parameter value at the highest Youden j was used as the threshold. The results are shown in Table 2, along with the sensitivity and specificity associated with each threshold, offering insight into the true positive and negative rates achieved by the individual parameters.

**Table 2 ijms-25-05107-t002:** The results of linear logistic regression analysis show the parameters sorted from best to worst predictor of poor prognosis.

Non-Diabetic Patients					Diabetic Patients				
Parameters	Threshold	Youden_j	Sensitivity (%)	Specificity (%)	Parameters	Threshold	Youden_j	Sensitivity (%)	Specificity (%)
D-Dimers	180	0.604	90	70.37	IL-18	206.1	0.398	58.82	80.95
eGFR	69.84	0.589	73.91	85	HDL	46	0.375	50	87.5
IL-6	36.27	0.581	63.64	94.44	Triglycerides	131	0.375	73.68	63.83
Creatinine	1	0.571	69.57	87.5	NETs	143020	0.362	60.61	75.61
Age	58.9	0.545	86.96	67.5	NLR	5.83	0.325	65.79	66.67
Potassium	4.5	0.48	60.87	87.18	Neutrophils	11.83	0.317	42.11	89.58
ESR	18	0.444	72.22	72.22	Fibrinogen	367	0.301	74.29	55.81
NLR	5.51	0.439	77.27	66.67	S100A8/A9	2197.97	0.296	67.65	61.9
Neutrophils	11.13	0.437	59.09	84.62	Lymphocytes	1.88	0.295	84.62	44.9
Ferritin	235	0.422	63.64	78.57	IL-1β	2.28	0.276	66.67	60.98
Phosphate	4.2	0.412	50	91.18	WBC	14.16	0.27	47.37	79.59
Fibrinogen	499	0.397	42.86	96.88	Potassium	3.9	0.229	92.31	30.61
CRP	17	0.35	52.17	82.86	Glycaemia	241	0.202	44.74	75.51
Glycaemia	108	0.329	73.91	58.97	Uric Acid	6.4	0.198	55.26	64.58
Lymphocytes	1.03	0.328	47.83	85	Troponin	66.4	0.187	37.84	80.85
IL-18	166.37	0.318	81.82	50	*Human* Klotho	75.91	0.186	42.42	76.19
WBC	13.19	0.315	56.52	75	HbA1c	7.3	0.182	48.65	69.57
LDL	136	0.314	73.91	57.5	Phosphate	3.4	0.181	80.65	37.5
Uric Acid	5.8	0.29	69.57	59.46	D-Dimers	220	0.18	62.96	55
Troponin	80	0.263	39.13	87.18	Magnesium	2.06	0.179	27.03	90.91
Gas6	21.33	0.239	27.27	96.67	PTH	45.21	0.179	46.88	71.05
*Human* Klotho	44.3	0.23	36.36	86.67	Gas6	28.92	0.176	17.65	100
Triglycerides	163	0.225	100	22.5	Age	52.3	0.173	94.87	22.45
HbA1c	5.5	0.214	72.73	48.65	LDL	112.8	0.151	56.76	58.33
Magnesium	2	0.21	69.57	51.43	Creatinine	0.9	0.147	56.41	58.33
Sodium	139	0.203	47.83	72.5	eGFR	60.41	0.136	28.21	85.42
PTH	65.72	0.159	38.1	77.78	ESR	57	0.122	25	87.18
Insulinemia	6.37	0.156	34.78	80.77	Insulinemia	6.78	0.114	90.91	20.51
Calcium	9	0.155	69.57	45.95	Ferritin	42.93	0.095	96.97	12.5
NETs	449510	0.081	95.24	12.9	Calcium	9.9	0.09	13.51	95.45
HDL	35	0.076	82.61	25	CRP	15.5	0.086	44.74	63.83
IL-1β	1.18	0.073	77.27	30	IL-6	23.83	0.083	50	58.33
S100A8/A9	4968.36	0.055	95.45	10	Sodium	130	0.051	5.13	100

For non-diabetic patients, the analysis revealed that the best predictors of unfavorable prognosis were D-Dimers, estimated glomerular filtration rate (eGFR) (unsurprisingly, also creatinine and age), potassium, ESR and NLR (Table 2). To further identify the best linear separation between patients with favorable versus unfavorable prognostic within the non-diabetic group, we performed a linear kernel SVM analysis based on D-Dimers, eGFR, Potassium, ESR, NLR, Ferritin, or D-Dimers and eGFR alone.

The results indicated that eGFR and D-Dimers exhibit comparable predictive efficacy as the larger set of six parameters that include them (area under the receiver operating characteristic (AUROC) 0.91 vs. 0.92, Figure 3A). This overlap in information arises from the observed correlations between certain markers under study, particularly within the group of inflammation-related markers (Figure 2A). Therefore, when we exclusively utilized eGFR and D-Dimers to predict prognosis (Figure 3B), the line best separating between favorable and unfavorable evolution had the equation:$$\log\left(eGFR\right)=1.037*\log\left(D-Dimers\right)+1.276$$

Since the slope of the line was very close to 1, for non-diabetic patients, we introduced a simplified prognostic score called the “N”-Score, which was computed as:N-Score=D-Dimeri/eGFR

For diabetic patients, the biomarkers that exhibited the highest predictive power were notably distinct from those found to be most influential in non-diabetic patients. Specifically, the top predictive biomarkers for diabetic patients included interleukin-18 (IL-18), high-density lipoprotein (HDL), Triglycerides, NETs, NLR, neutrophils, fibrinogen, and S100A8/9, as detailed in Table 2. The same SVM analysis (as for non-diabetics) was performed for these biomarkers (without neutrophils and lymphocytes, included in the analysis as NLR) or for two sub-sets of two parameters: IL-18/HDL and NETs/IL-18. The results revealed that NETs/IL-18 provided similar predictive power to the set of 8 parameters comprising them (AUROC 0.7 vs. 0.72, Figure 3C) and a better predictive power when compared to the IL-18/HDL combination (AUROC 0.66). In the case of the NETs/IL-18 scatterplot (Figure 3D), the line best separating between favorable and unfavorable prognosis within the diabetic group had the equation:$$\log\left(IL18\right)=-0.537*\log\left(NETs\right)+11.62$$

Since the slope of the right is close to −0.5, the prognostic score for diabetic patients was computed as:D-Score=IL-18∗sqrt(NETs)

### 2.4. Neutrophils Isolated from ACS Patients Exhibit a Pro-Inflammatory Gene Expression Pattern

As clinical data revealed that unfavorable evolution post-MI was associated with increased inflammatory status, we further characterized the inflammatory state of neutrophils isolated from several of the MI patients included in this study. The expression levels of different inflammatory genes were evaluated in neutrophils isolated from peripheral blood in the first 24 h post-MI. As shown in Figure 4A–F, the neutrophils isolated from MI patients with negative prognosis (MI_NP) exhibited increased gene expression of almost all analyzed pro-inflammatory mediators: *CCL3, IL-1β, IL-18, S100A9*, and of *the intracellular cell adhesion molecule-1* (*ICAM-1)*, which is known to mediate neutrophils infiltration in the infarcted area [13]. In addition, the *p22phox, Nox2*, and *matrix metalloprotease (MMP-9)* also presented an increased gene expression, suggesting an increased oxidative status and remodeling capacity of neutrophils from MI patients with negative prognosis (MI_NP) (Figure 4G–I), which can account for disease aggravation.

When analyzing the expression of the same molecules in neutrophils from ACS patients with diabetes compared to non-diabetic individuals, regardless of their prognosis, the results revealed no significant differences in the gene expression levels of pro-inflammatory molecules *CCL3, IL-1β, IL-18, IL-6, S100A9, ICAM-1* (Figure 5A–G), data in agreement with clinical evaluation (Figure 5D–F). However, the *p22 subunit* of Nicotinamide adenine dinucleotide phosphate (NADPH) oxidase and *MMP-9* expression significantly increased in neutrophils from diabetic patients, as compared with non-diabetic patients (Figure 5H,I). These data suggest that in the acute phase, while the differences regarding inflammation cannot be observed in neutrophils from diabetic versus non-diabetic patients, significant differences can still be found regarding oxidative and remodeling molecules.

### 2.5. Protein Expression of Pro-Inflammatory Molecules in Neutrophils Isolated from MI Patients with or without Negative Prognosis

As the proteins are the functional, active effectors, the expression of some of the pro-inflammatory molecules found modified at the gene expression levels in neutrophils from MI patients with negative prognosis (NP) were further investigated at the protein expression level. The Western Blot results corroborated well with the qPCR data, showing that protein expression of pro-inflammatory molecules, S100A9, monocyte chemoattractant molecule-1 (MCP-1), and IL-1β were significantly higher in neutrophils from patients with negative prognosis (Figure 6A–C). In addition, myeloid differentiation factor 88 (MyD88—a molecule adaptor that is critical for innate immune function and plays a role in neutrophil recruitment and myocardial injury after transient ischemia [14], was significantly increased in neutrophils from NP patients (Figure 6D). Similarly, the protein expression of NADPH subunit p22phox and MMP-9 gelatinase was found elevated in the lysate of neutrophils from MI_NP versus MI patients (Figure 6E,F), confirming the gene expression data.

### 2.6. NETs Are Increased in Serum and Neutrophils of MI Patients and Are Accompanied by Increased Expression and Activity of Neutrophil Elastase

In addition to inflammatory molecules, neutrophil activation and degranulation also imply the formation of NETs and the release of different enzymes, including neutrophil elastase (NE) and myeloperoxidase (MPO), with key roles in both host defense [15] and disease [16,17]. In vascular disorders, NETs have pro-thrombotic properties by stimulating fibrin deposition. Moreover, increased NET levels correlate with larger infarct size and predict major adverse cardiovascular events [18]. Therefore, we quantified the NET level in serum from MI patients versus healthy controls and found significantly increased NETs in the serum of MI patients (Figure 7A). Moreover, the NETs released by neutrophils from MI patients or healthy subjects were highlighted by fluorescence microscopy. After 2 h in culture, neutrophils from MI patients exhibited increased NET structures compared with neutrophils from healthy controls (Figure 7B).

Regarding neutrophil-specific enzymes, our data shows that protein expression of NE, but not of MPO, is significantly increased in neutrophils from MI_NP, as compared with MI patients (Figure 7C,D). As NE was different between the groups, we further investigated the enzymatic activity of NE in the secretome of neutrophils from patients and from controls, and the results show that NE activity is increased in neutrophil media of ACS patients, compared with voluntary controls. Moreover, NE activity in secretome released by neutrophils from patients with MI _NP was higher compared with that of neutrophils from MI patients (Figure 7E). Importantly, differences were also found between diabetic and non-diabetic MI patients, with elastase activity levels elevated in the secretome of neutrophils from diabetic patients (Figure 7F).

The localization of MPO and elastase in neutrophils from healthy control or MI patients was investigated by immunofluorescence (Figure 8). The results show that in neutrophils isolated from control subjects, both enzymes exhibited granular staining, localized in the cytoplasm of cells, surrounding the neutrophil nucleus (Figure 8a,b). In neutrophils isolated from MI patients, in the first 24 h post-MI, MPO remained granular or co-localized with DNA in neutrophils undergoing NET release (arrows, Figure 8b and inset 8b1), and elastase exhibited reduced granular staining and co-localize with nucleus (Figure 8d and inset 8d1).

## 3. Discussion

Inflammation represents an intricate and highly regulated series of molecular and cellular processes, currently acknowledged as a modifiable factor contributing to cardiovascular risk. Among the various factors linked to inflammation, several studies have underscored the significance of differentials in white blood cell counts, with particular emphasis on neutrophils, lymphocytes, and monocytes. Following MI, there is an increased release of neutrophils into the peripheral blood, resulting in neutrophilia. Elevated neutrophil counts in blood have the potential to predict both acute and chronic cardiovascular outcomes [19,20]. Furthermore, an elevated NLR has been correlated not only with CVD but also with short-term adverse cardiovascular events, including mortality, coronary artery disease, stroke, and heart failure [21]. In view of recent reports regarding the existence of different populations of neutrophils with distinct functionalities, it is, therefore, possible that disease evolution post-MI is associated with mobilization of neutrophils with altered phenotype, which might contribute to ACS risks.

Our study shows an increased inflammatory status in MI patients with unfavorable evolution and found that neutrophil activity had important prognosis value for diabetic patients. The clinical data are sustained by in vitro results showing that neutrophils from MI patients with unfavorable evolution exhibit increased expression of pro-inflammatory molecules *CCL3*, *IL-1β*, *IL-18*, *S100A9*, and NETs, but also of molecules with essential roles in adverse progression post-MI: *p22phox*, *Nox2*, and metalloprotease *MMP-9*. Therefore, this study demonstrates a substantial increase in the inflammatory status of MI patients with an unfavorable evolution, which can be attributed to changes in the phenotype of neutrophils.

Previous data demonstrated that acute MI is associated with significant systemic inflammatory response. When coupled with pre-existing chronic inflammation (as in diabetic patients), it can prolong the process of myocardial healing and potentially could result in the formation of a deficient scar and/or adverse ventricular remodeling [22]. Our data support and extend these findings, showing that neutrophil count, NLR, NETs, and S100A8/A9, related to neutrophil activation, were increased in the MI group with unfavorable evolution. Neutrophils promote and maintain the inflammatory response by synthesizing other cytokines like IL-1, IL-6, IL-18, IL-12, and TGF-β, which in turn attract and reactivate the neutrophil, creating an inflammatory loop [23,24]. We also found that CRP, IL-6, IL-1β and ESR were increased in the negative evolution MI group. In addition, we determined strong correlations between inflammatory markers that could be related to neutrophil activation (Figure 2A—matrix correlation). Overexpressed inflammatory molecules, along with granular proteins such as MPO, NE, and MMPs released by activated neutrophils, affect not only the extracellular matrix but also act like myocardial depressants that negatively affect myocyte contractility [23,24]. Moreover, NETs and decreased deformability of activated neutrophils contribute to microvascular obstructions, no re-flow, and cardiac reperfusion injury [25]. In accordance with these data, we found that neutrophil number and NLR were increased in the unfavorable group and correlated negatively with LVEF. On the same note, troponin expressing myocardial necrosis extent was positively correlated with white blood cells, neutrophils, NLR, S100A8/A9, NETS, and IL-6, strengthening the conclusion that the activated neutrophils contribute to myocyte dysfunction after myocardial infarction.

Acute and chronic hyperglycemia have been demonstrated to amplify the pro-inflammatory processes and to negatively impact the innate immune system [26]. In the diabetic milieu, neutrophils lack the weapons that protect the host against pathogens, like chemotaxis, phagocytosis and intracellular ROS, but they overexpress the inflammatory pathways like NET formation and extracellular ROS and augment cytokine expression [27]. We found that diabetics have poorer prognoses and distinct parameters that influence in-hospital evolution after MI when compared to non-diabetics. Most of these influential factors are related to inflammation and are a consequence of neutrophil activation, encompassing variables such as neutrophil count, NETs, NLR, and S100A8/A9. Interestingly, the absolute values of these parameters were not significantly different between diabetics and non-diabetics, but when evaluating prognosis, the high values of these biomarkers were more predictive of poor evolution in diabetic patients. Consequently, a prognostic score with a specificity and sensitivity of 85% created for non-diabetic patients had no predictive value in the evolution of diabetic patients; however, a prognostic score created for diabetic patients (overall less sensitive and specific) still provided some predictive value in non-diabetic patients. These results suggest that the molecular mechanisms and ensuing disease progression may be distinct in diabetic versus non-diabetic patients with MI and should raise awareness of clinicians in evaluating distinct parameters when predicting evolution and complications.

Interesting results that remained in the clinical data arose from the in vitro analysis of circulating neutrophils, showing an altered phenotype of neutrophils from MI patients. Hence, when comparing neutrophils from healthy subjects with those from MI patients, all the inflammatory molecules were significantly elevated in neutrophils from MI patients (Appendix A). Notably, neutrophils from MI patients with unfavorable evolution exhibit a significant upregulation of inflammatory cytokines such as *CCL3, IL-1β, IL-18*, and alarmin *S100A9* compared with MI patients with a favorable evolution. Our recent transcriptomic analysis revealed that all these molecules define the pro-inflammatory subtype of neutrophils [28], and these data suggest that the circulating neutrophils of these patients are committed to the pro-inflammatory phenotype before reaching the infarct zone. Intriguingly, when comparing the gene expression of these same molecules between neutrophils from MI patients with or without diabetes, these distinctions disappeared. This suggests that the acute-phase post-MI inflammatory cascade masks the specific effects induced by hyperglycemia, rendering them indistinguishable in this context. The observed increase in the expression of pro-inflammatory cytokines among patients with an unfavorable disease progression aligns with prior research findings, showing significant upregulation of intramyocardial cytokines, including *TNF-α* and *IL-1β* mRNA expression, in the infarcted area [29]. Notably, this upregulation of cytokines can revert to baseline levels in cases of small infarctions, or it can be sustained or even give rise to a secondary wave of upregulation, particularly in instances of large infarctions or when the host’s inflammatory response is excessively heightened, ultimately contributing to the chronic remodeling phase [30]. The elevation in cytokine expression is usually accompanied by increased ROS and precedes the consequent increase in local matrix MMP activity in the infarct area. Our data indicate a heightened expression of the oxidative stress molecule p22phox and of MMP-9 within circulating neutrophils of MI patients who experience unfavorable disease progression. This elevation potentially serves as an additional source for the generation of ROS and for the subsequent degradation of the extracellular matrix upon recruitment of these neutrophils to the infarcted area. Notably, these two molecules, which have a substantial impact on disease progression [31,32], continue to exhibit significant increases in diabetic MI patients when compared to their non-diabetic counterparts. This persistence can augment the oxidative stress and the matrix remodeling when reaching the infarcted area, and may well contribute to the exacerbated disease evolution observed in diabetic patients.

This altered profile of neutrophils was, in addition, accompanied by an exacerbated level of NETs found both in the serum or neutrophils from MI patients, compared with healthy controls. It is known that upon activation, during the formation of NETs, neutrophil elastase translocates from the granules to the nucleus via an unknown mechanism that requires ROS [15], where it partially degrades specific histones, promoting chromatin decondensation. Subsequently, myeloperoxidase synergizes with NE in driving chromatin decondensation independent of its enzymatic activity [15]. Looking at the protein expression of these two granular molecules, we found significantly increased protein levels of NE, but not of myeloperoxidase, in neutrophils from MI patients with unfavorable evolution. Moreover, the activity of NE was found to be elevated in the serum of these patients and of MI patients with diabetes, indicating the activation status of neutrophils. Previously, plasma NE was also found to be increased in the diabetic group of patients [33], and inhibition of NE improved myocardial dysfunction and cardiac survival by upregulating insulin/Akt signaling post-MI [34]. Therefore, increased expression and activity of NE by neutrophils from MI patients with unfavorable outcomes or diabetes could represent a contributing factor to the adverse post-MI progression in these individuals.

Concerning the mechanisms that connect unfavorable disease progression with heightened inflammation, one plausible factor could involve the differential modulation of cell surface receptors responsible for transmitting signals from the external environment to the interior of the cell. In connection with this aspect, our data found that neutrophils from MI patients with adverse outcomes exhibited elevated levels of *toll-like receptor-4* (*TLR4)*, as indicated in the Appendix A. TLR4 is a well-known receptor that plays a significant role in mediating maladaptive remodeling and impairing cardiac function following myocardial infarction [35].

## 4. Materials and Methods

### 4.1. Study Participants

We conducted a longitudinal prospective study of a total of 229 patients hospitalized between April 2021 and September 2022 in “Elias” Emergency Hospital Bucharest, Romania. A total of 63 were non-diabetics, 78 had pre-diabetes, and 88 were diabetics (Table 1). As blood sugar is elevated in the acute state of MI, we used glycated hemoglobin and previous history to define the diabetic and pre-diabetic status according to the ADA 2019 definition [36]. To assess clinical factors and biomarkers with prognosis value, the group was divided into a subgroup with unfavorable evolution (with in-hospital complications, n = 77 patients) and a subgroup with favorable evolution (without complications, n = 152). Unfavorable evolution was defined as the presence of Killip stage 3 or 4 on admission, mechanical complications, significant ventricular arrhythmia (VT, VF or with hemodynamic impairment), discharge LVEF < 40% or in-hospital death. To mitigate the potential impact of severely reduced ejection fractions on our primary goal of evaluating the predictive significance of inflammation, we excluded patients with LVEF below 30% from the study. When assessing prognostic factors, we included patients with LVEF between 30% and 40% in the criteria for poor outcomes.

Inclusion criteria were as follows: patients within the first 24 h of acute myocardial infarction with ST-segment elevation according to the universal MI definition [37] with a left ventricular ejection fraction higher than 30% at presentation. We decided to limit enrollment of patients with ACS and severely depressed LVEF <30% due to the well-known influence on the outcome [38,39] and the independent effect on inflammatory markers [40,41] irrespective of the underlying cause. Severely depressed LVEF may be a source of bias that can significantly change inflammatory markers in the study group. Therefore, we considered an LVEF between 30 and 40% as a criterion for acute complications of the ACS without having a major effect on inflammatory markers, despite prognostic consequences. Patients with acute viral, bacterial, or fungal infections, those with known autoimmune diseases, or severe pre-existing organ dysfunctions were excluded. Written informed consent was obtained from participants in accordance with the Declaration of Helsinki. The study protocol was approved by the Ethics Committee of “Elias” Emergency Hospital, Bucharest, Romania (no 3349/06.05.2021).

A group of healthy volunteers (*n* = 10) who did not have cardiovascular or any chronic disease was included as a reference for some of the investigated molecules/parameters.

### 4.2. Clinical Evaluations

Physical examination was performed daily starting from the moment of emergency room admission. Traditional risk factors for both diabetes and atherosclerotic disease, such as age, sex, smoking, hypertension, dyslipidemia, and body mass index, were assessed. Medical history and previous treatment were documented for each patient.

Investigations: Blood samples according to standard care for MI were drawn and analyzed on admission, during the hospitalization, and at discharge. Inflammation markers such as CRP, ferritin, ESR, and specific diabetic markers such as glycated hemoglobin and insulinemia were also evaluated during hospital stay. The usual biological parameters were measured according to the hospital’s standard laboratory procedures.

Twelve lead electrocardiograms were evaluated daily. Cardiac rhythm was continuously monitored in the Coronary Care Unit for the first 72 h post-MI for arrhythmias. Transthoracic echocardiograms were performed at admission and discharge or in case of new symptoms. LVEF was assessed using the 2D biplane method, according to the modified Simpson’s rule. LVEF dynamics were monitored. During the hospital stay, the images and echo interpretation were assessed by at least two experienced cardiologists. Angiogram and interventional revascularization were performed at inclusion in the first 60 min after admission in the ER. The culprit lesion, the number of main coronary vessels that presented with a stenosis higher than 50%, and the total number of coronary stenoses were documented.

### 4.3. ELISA Assay

Peripheral blood samples were collected in sterile serum tubes in the first 12 to 24 h after the onset of MI symptoms, and serum was isolated by centrifugation (15 min at 1500× *g*). The serum was stored at –80 °C and further used for quantification of inflammatory markers by ELISA assay using specific kits from R&D Systems (Minneapolis, MN, USA): IL-1β (DY201), S100A8/A9 Heterodimer (DY8226-05), IL-18 (DY8936), Klotho (DY5334), and Gas6 (DY885B) following the manufacturer’s instructions.

### 4.4. SYTOX Green Assay

An amount of 50 μL of 10× diluted serum samples or 50 μL conditioned media of neutrophils from healthy controls and MI patients exposed for 1 h to 50 mM phorbol 12-myristate 13-acetate (PMA) were added to a black 96-well microplate. SYTOX Green fluorescent dye at a concentration of 5 μM (#S33025, Invitrogen, Carlsbad, CA, USA)) was added to each well. The plate was incubated in the dark for 5 min at RT, and fluorescence was read on a Tecan Microplate Reader Infinite^®^ 200 Pro, with excitation and emission wavelengths of 485/527 nm.

### 4.5. Cells

Peripheral blood samples from 35 patients with ACS and those without diabetes were collected in Vacutainer EDTA blood collection tubes. After blood collection, the neutrophils were isolated by gradient centrifugation using Polymorphoprep (Proteogenix, Miami, FL, USA) as we previously described [28] or using a *human* neutrophil isolation kit (Miltenyi, Teterow, Germany) according to manufacturer instructions. After isolations, neutrophils were subjected to total RNA or protein isolation or were used for different analyses as described below.

### 4.6. Quantitative RT-PCR

Total RNA was isolated from neutrophils using TRIzol (Invitrogen), and first-strand cDNA synthesis was performed employing 1 μg of total RNA and MMLV reverse transcriptase, according to the manufacturer’s protocol (Invitrogen). Assessment of mRNA expression of different pro-inflammatory molecules was done by amplification of cDNA using a LightCycler 480 Real-Time PCR System (Roche, Basel, Switzerland) and SYBR Green I. The primer sequences for the mRNAs of interest are shown in Appendix A. The relative quantification was done using the comparative CT method and expressed as arbitrary units. *Beta-2* microglobulin was used as a reference gene.

### 4.7. Western Blot

Following isolation, neutrophils were pelleted, and the cell pellets were lysed using RIPA lysis buffer supplemented with a protease inhibitor cocktail. After centrifugation (12,000× *g*), the proteins were quantified by bicinchoninic acid (BCA) Protein Assay Kit. Samples of 30 proteins were separated on 10% or 12% SDS-PAGE (sodium dodecyl sulfate-polyacrylamide) gel electrophoresis and transferred to nitrocellulose membranes. The resulting blots were subsequently probed with specific primary antibodies directed against IL-1β (MAB601, R&D Systems), S100A9 (PA1-46489, ThermoFisher, Waltham, MA, USA), MCP-1 (ab9669, Abcam), MyD88 (AF2928, R&D Systems), p22phox (SC-271262, Santa-Cruz Biochtenology, Dallas, TX, USA), MMP-9 (MAB936, R&D Systems), Neutrophil Elastase (MAB91671-100, R&D Systems), MPO (AF3667, R&D Systems), GAPDH (ab9485, Abcam, Cambridge, UK), and β-tubulin (ab6046, Abcam). These were followed by secondary antibodies: anti-rabbit HRP-coupled (HAF008, R&D Systems), anti-*mouse* HRP-coupled (#31430, R&D Systems), or anti-goat HRP-coupled (HAF017, R&D Systems). The signals were visualized using SuperSignal West Pico chemiluminescent substrate (Pierce, Appleton, WI, USA) and quantified by densitometry employing a gel analyzer system, a LAS 4000 luminescent image analyzer (Fujifilm, Minato City, Tokyo, Japan), and image reader LAS 4000 software.

### 4.8. Fluorescence Microscopy of NET Formation

Freshly isolated neutrophils from MI patients or health subjects (1 × 10^6^ cells/mL) were seeded onto poly-L-lysine-coated wells in RPMI without FCS for 2 h in the presence of 2.5 µM SYTOX green. After 2 h, the medium was aspirated, cells were washed (twice), and Hoechst 33,342 (2 µM) was added for 20 min. After two washes, fluorescence microscopy was conducted with an Olympus IX81 microscope (Shinjuku City, Tokyo, Japan) equipped with an XM10 camera.

### 4.9. Detection of Neutrophil Elastase (NE) Activity

Following isolation, neutrophils from patients were exposed to PMA (50 nM) for 1 h, and the conditioned media with factors released by activated neutrophils (secretome) was used for NE quantification using the EnzChek^®^ Elastase Assay Kit (Invitrogen, Carlsbad, CA, USA, Catalog No. E12056), according to the manufacturer’s instructions.

### 4.10. Cell Immunofluorescence

Neutrophils (1 × 10^6^ cells) in the RPMI 1640 medium with 2% FBS were cultured for 4 h on poly-L-lysine-coated coverslips in the presence of 2.5 µM SYTOX green. Subsequently, the cells were fixed in 4% paraformaldehyde for 20 min; the supernatant was removed gently, and the cells were washed twice with PBS. Cells were permeabilized with 0.3% triton X for 10 min and blocked with 2% BSA for 1 h, followed by incubation with anti-MPO *human/mouse* polyclonal antibody (R&D System, #AF3667, 2:100), or anti-*human* neutrophil elastase (R&D System, #MAB91671, 1:100) overnight at 4 °C. The cells were washed with PBS and incubated with secondary antibodies coupled with fluorophores: donkey anti-goat Alexa Fluor 594 antibody (Thermo Fisher Scientific, #A-11058, 1:400) or chicken anti-*mouse* Alexa Fluor™ 594 (Thermo Fisher Scientific, #A-11058, 1:400) for 1 h at RT in the dark. Cells were then incubated with DAPI for 10 min and were visualized with an Olympus IX81 inverted microscope (Shinjuku City, Tokyo, Japan).

### 4.11. Data Analysis and Statistics

Data were collected in IBM SPSS Statistic 20, and statistical analysis and visualizations were performed in Python 3 using the pandas v0.25.1 package [42] for database manipulation and matplotlib v3.1.1 packages [43] and seaborn v0.9.1.dev0 [44] for visualizations.

Each individual parameter was checked if normally or log-normal distributed using the Shapiro–Wilk test (SciPy v1.3.1 package [45]) and logarithmized to base 10 if needed.

To assess correlations between parameters, the Pearson R coefficient and the corresponding *p*-value were calculated individually for each pair of variables using SciPy. The results were then compiled into a matrix, the columns and rows were grouped by the unweighted pair group method with the arithmetic mean method and plotted using the seaborn v0.9.1.dev0 package. For each numerical parameter measured, differences between groups (diabetic, pre-diabetic, and non-diabetic) were checked using one-way ANOVA analysis with Tukey’s post hoc test for multiple comparisons. Alternatively, Student *t*-tests (SciPy) were used for comparisons between two groups (favorable prognosis vs. unfavorable prognosis).

Categorical variables were used to construct contingency tables indicating the frequency of discrete values in the study groups. For the binary values, the statistical analysis was done using the Fisher exact test analysis, and for the variables that could take more than two values, the chi-square analysis (SciPy) was used. A linear logistic regression model (sklearn v0.21.3, [46]) was used to evaluate the prognostic value of different parameters. Biomarker sensitivity and specificity were calculated by receiver operating characteristic (ROC) analysis, and the threshold was selected to correspond to the highest Youden J statistic (sensitivity + specificity −1). For marker combinations, a support vector machine (SVM) model with a linear kernel (sklearn v0.21.3package) was used.

GraphPad Prism 7.0 with data points expressed as mean ± standard deviation (SD) was used for all statistical analyses of in vitro results. We used a two-tailed Student’s *t*-test when comparing two experimental groups and a one-way ANOVA and Tukey’s multiple comparison test when comparing more than two groups. A *p*-value of *p* < 0.05 was considered statistically significant.

## 5. Conclusions

Overall, our data present evidence that poorer in-hospital evolution of MI patients is related to systemic inflammatory markers and that neutrophil-related biomarkers are more specific when predicting short-term prognosis in diabetics. Importantly, inflammation has a short-term prognostic role in acute myocardial infarction, not only in patients with severely reduced ejection fraction. Thus, our research showed that inflammatory markers are associated with poor short-term outcomes after ACS in patients with an LVEF lower than 40%. The effect of systemic inflammatory response on the mechanical performance of the myocardium, both systolic and/or diastolic, remains to be further investigated.

Circulating neutrophils from ACS patients exhibit an altered, pro-inflammatory phenotype before reaching the infarct zone, which is exacerbated in patients with poor prognosis. The notable disparities in inflammatory profiles observed in neutrophils from MI patients with a complicated in-hospital evolution imply potential modifications in the innate immune response within this specific patient group, potentially exerting detrimental effects on the subsequent course of the disease.

## Figures and Tables

**Figure 1 ijms-25-05107-f001:**
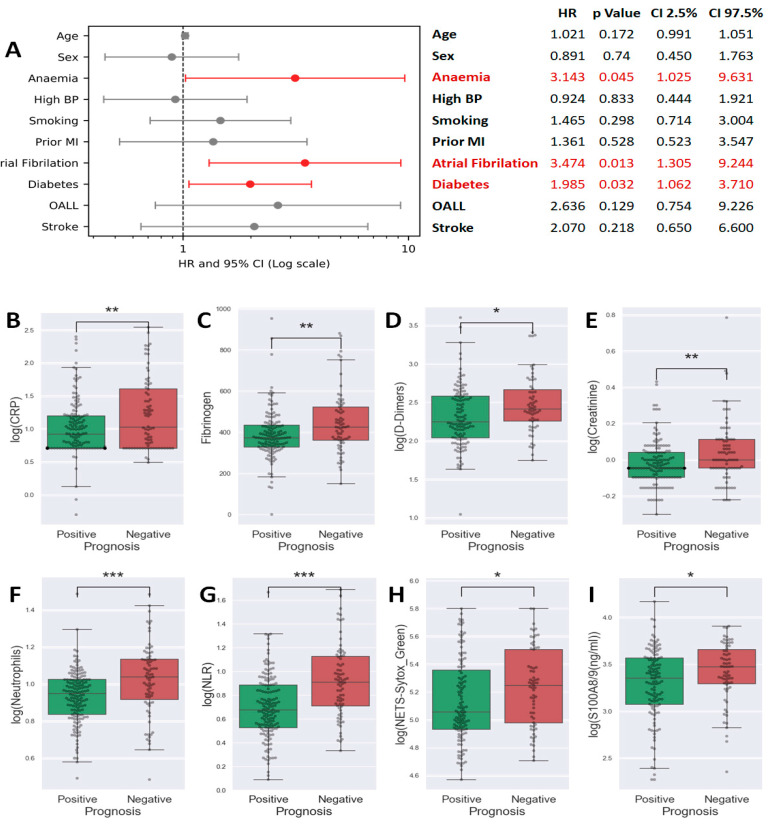
Forest plot of a multinomial logistic regression analysis using the stats models package in Python 3. Data were plotted using matplotlib as an Odds Ratio plus error bars for the 95% confidence interval (from 2.5% to 97.5%). Statistically significant results were represented in red (**A**). Box plots showing the comparison of C-reactive protein (CRP) (**B**), Fibrinogen (**C**), D-Dimers (**D**), Creatinine (**E**), Neutrophils (**F**), neutrophil-to-lymphocyte ratio (NLR) (**G**), neutrophil extracellular traps (NETs) (**H**), and S100A8/9 (**I**) levels between patients with favorable vs. unfavorable prognosis (Student *t*-test, * *p* < 0.05, ** *p* < 0.01, *** *p* < 0.001).

**Figure 2 ijms-25-05107-f002:**
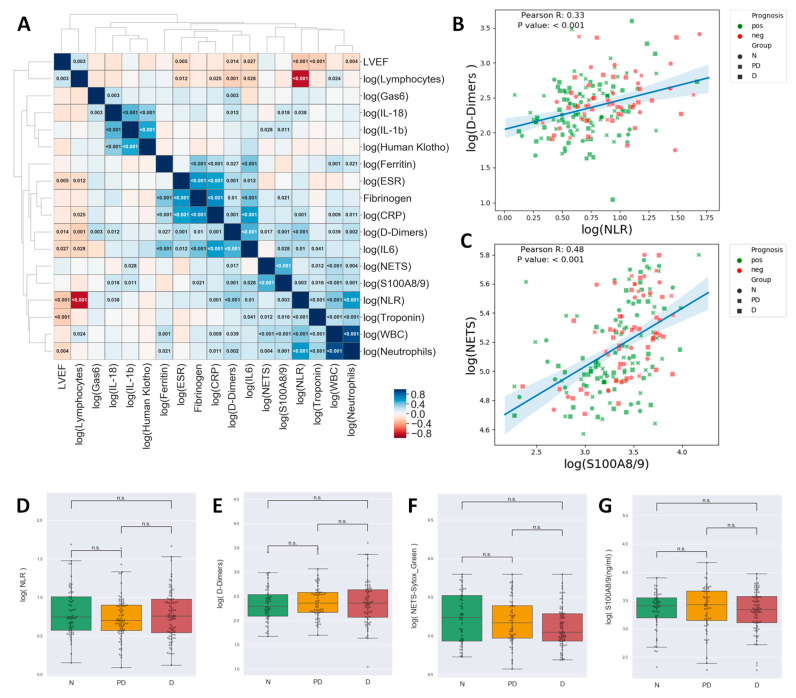
Correlation matrix for inflammatory markers. The cell colors indicate the Pearson R values, and for the significant correlations, the values indicated in the cells represent the *p*-value. Clustering of the parameters was performed by the unweighted pair group method with arithmetic mean and led to the identification of three highly intercorrelated clusters plus lymphocytes who tended to negatively correlate with most other markers (**A**) Scatterplots showing the correlation between NLR and D-Dimers (**B**) and NETs and S100A8/9 (**C**). Notice that High NETs/High S100A8/9, High D-Dimers/High NLR tend to be associated with unfavorable prognosis. Pearson R and *p*-values are indicated in the plots. Box plots showing the comparison of NLR (**D**), D-Dimers (**E**), NETs (**F**), and S100A8/9 (**G**) between Non-diabetic (N), Pre-Diabetic (PD), and Diabetic (D) patients. (one-way ANOVA, Tukey’s post hoc test, n.s.—not significant).

**Figure 3 ijms-25-05107-f003:**
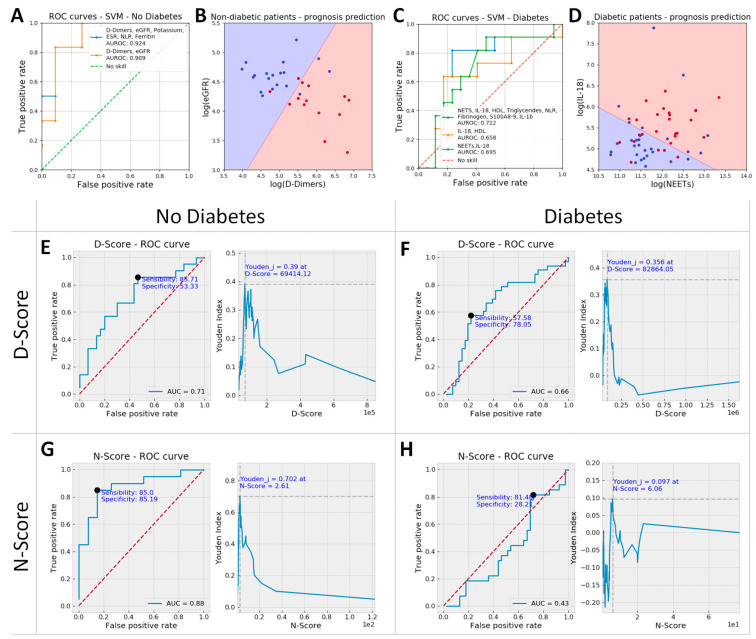
ROC curves for the top performing parameters: D-Dimers, estimated glomerular filtration rate (eGFR), Potassium, erythrocyte sedimentation rate (ESR), NLR and Ferritin vs. only D-Dimers and eGFR for non-diabetic patients (**A**), and NETs, interleukin-18 (IL-18), high-density lipoprotein (HDL), triglycerides, NLR, fibrinogen, S100A8/9 and interleukin-1β (IL-1β) vs. IL-18 and HDL or IL-18 and NETs for diabetic patients (**C**). D-Dimers and eGFR were later used to calculate N-Score and IL-18 and NETs used for the D-Score, based on the equation of the boundary line in the scatter plots shown in (**B**) and (**D**), respectively. These scatter plots show data from patients with unfavorable evolution as red dots and data from patients with favorable evolution as blue dots, while the decision space is shown as background color (red: unfavorable and blue: favorable evolution). Thus, red dots on a red background and blue dots on a blue background refer patients with correct evolution prediction (true positives and true negatives, respectively); red dots on a blue background show false negative results, and blue dots on a red background show false positive results in predicting unfavorable evolution. The N-score showed for all non-diabetic patients a sensitivity and specificity of 85% each when a prognostic threshold value of 2.61 was set (area under the receiver operating characteristic [AUROC] = 0.88, **G**), and the D-score had a poorer prognostic value, showing a sensitivity of 57.6% and a specificity of 78.1% with a threshold value of 8286 for all diabetic patients in the study (AUROC = 0.66, **F**). While D-Score had prognostic value even when used in non-diabetic patients (AUROC = 0.71, **E**), the N-score is completely devoid of predictive value in diabetic patients (**H**).

**Figure 4 ijms-25-05107-f004:**
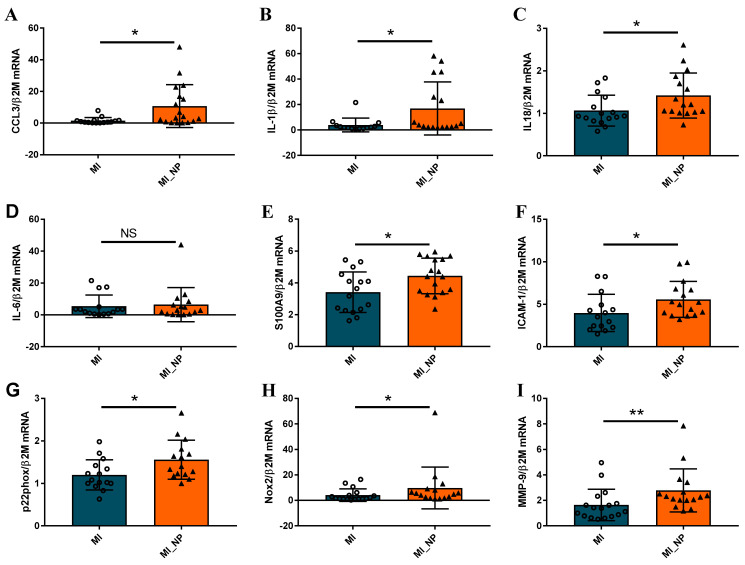
Neutrophils from patients with myocardial infarction and negative prognosis (MI_NP) exhibit increased expression of pro-inflammatory genes. (**A**–**I**) The expression of genes associated with an inflammatory phenotype of neutrophils *CCL3*, *IL-1β*, *IL-18*, *IL-6*, *S100A9*, *intracellular cell adhesion molecule-1 (ICAM-1)*, *p22phox*, *Nox-2*, and *matrix metalloprotease MMP-9* were investigated in neutrophils from patients with ACS (MI) or patients with ACS and negative prognosis (MI_NP). * *p* < 0.05, ** *p* < 0.01, (MI_NP vs. MI), NS—not significant.

**Figure 5 ijms-25-05107-f005:**
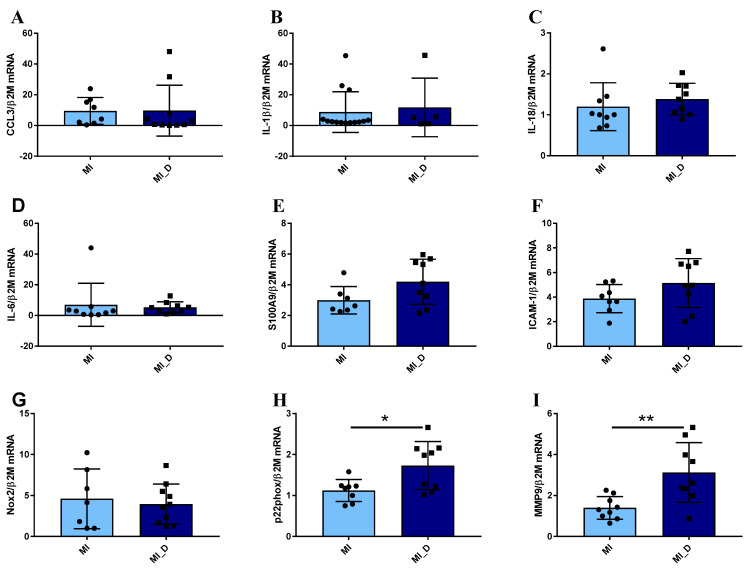
Gene expression of pro-inflammatory molecules in neutrophils isolated from ACS patients with or without diabetes. (**A**–**I**) qPCR analysis of pro-inflammatory markers *CCL3*, *IL-1β*, *IL-18*, *IL-6*, *S100A9*, and *ICAM-1*, showed no significant differences between neutrophils from MI patients with diabetes (MI_D) versus MI without diabetes (MI). In contrast, the *p22phox* and *MMP-9* expression significantly increases in neutrophils from diabetic patients (**H**–**I**). * *p* < 0.05, ** *p* < 0.01 (MI_D versus MI).

**Figure 6 ijms-25-05107-f006:**
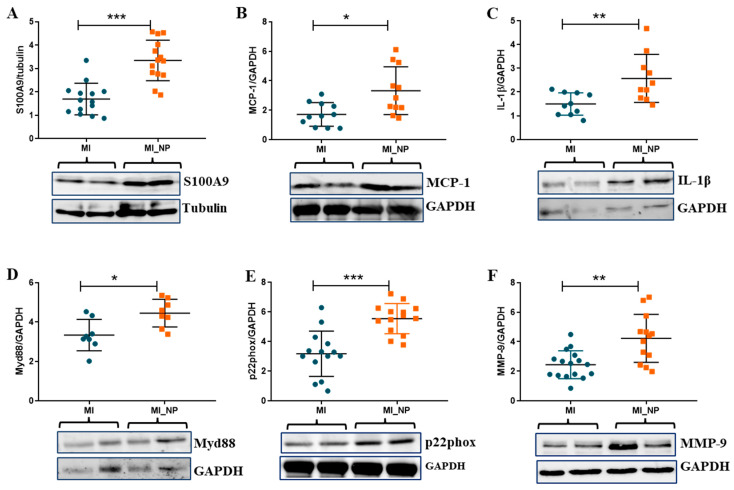
Protein expression of pro-inflammatory molecules expressed by neutrophils isolated from ACS patients with or without negative prognosis. (**A**–**D**). Quantification of protein expression and representative images of pro-inflammatory molecules S100A9, monocyte chemoattractant molecule-1 (MCP-1), IL-1β, and myeloid differentiation factor 88 (Myd88) as determined by Western Blot. (**E**,**F**) Quantification of protein expression and representative images of p22phox and MMP-9 as determined by Western Blot. * *p* < 0.05, ** *p* < 0.01, *** *p* < 0.001 (MI patients versus MI patients with negative prognosis—MI_NP).

**Figure 7 ijms-25-05107-f007:**
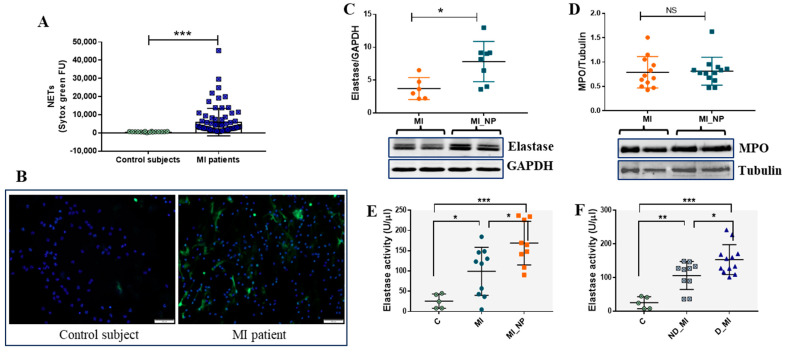
Evaluation of NETs and of neutrophil enzymes elastase and myeloperoxidase (MPO) in neutrophils from MI patients. (**A**) NET levels in serum from healthy controls and MI patients, as detected by SYTOX green. (**B**) NETs staining in neutrophils from healthy control and MI a patient, after 2 h in culture as stained by SYTOX green (green). Nuclei are stained with DAPI (blue). The image scale size is 100 µm. (**C**,**D**) Protein expression of elastase (C) and myeloperoxidase (D) in neutrophils from MI patients with negative prognosis (MI_NP) versus MI patients. Neutrophils isolated from MI or MI_NP were immediately lysed and investigated by Western blot. (**E**,**F**) Elastase activity in the secretome of phorbol 12-myristate 13-acetate (PMA)-exposed neutrophils from healthy voluntaries (C), MI patients (MI), MI patients with negative prognostic (MI_NP), non-diabetic patients with MI (ND_MI), and diabetic patients with MI (D_MI), using a commercial kit (Invitrogen, Carlsbad, CA, USA), * *p* < 0.05, ** *p* < 0.01, *** *p* < 0.001.

**Figure 8 ijms-25-05107-f008:**
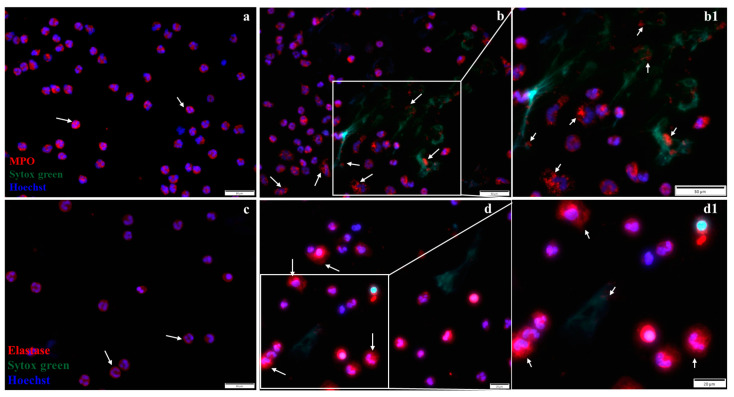
Immunofluorescent staining of MPO and elastase. (**a,b**) Representative fluorescence images of MPO localization in neutrophils from a healthy subject (**a**) and in neutrophils from an MI patient (**b**), stained with antibodies against MPO (red), NETs (green), and nucleus (DAPI-blue). The arrows indicate localization of MPO within the cytoplasm in neutrophils from a healthy subject (**a**) and in the cytoplasm or bound to NETs in neutrophils from an MI patient (**b**,**b1**). (**c,d)** Representative fluorescence images of elastase stained with red, localized in the cytoplasm of neutrophils from a healthy subject (**c**), and in the cytoplasm or bound to NETs in neutrophils from an MI patient (**d**,**d1**), as indicated by arrows. The scale size is 50 µm for (**a**,**b**,**b1**,**c**) images and 20 µm for images d and d1. The b1 and d1 images show inset regions of interest as marked by the white boxes.

**Table 1 ijms-25-05107-t001:** Population characteristics.

	Control (N)	Pre-Diabetes (PD)	Diabetes (D)	All Patients
Number	63	78	88	229
Sex				
F	19 (30.2%)	27 (34.6%)	29 (33%)	75 (32.8%)
M	44 (69.8%)	51 (65.4%)	59 (67%)	154 (67.2%)
Age ^#^	60.9 (40.1–82.1)	61.05 (46–79.3)	63.5 (43.8–79.9)	61.9 (41.5–81.4)
High Blood Pressure	40 (63.5%)	53 (67.9%)	74 (84.1%)	167 (72.9%)
Smokers	48 (76.2%)	47 (60.3%)	55 (62.5%)	150 (65.5%)
Anaemia	5 (7.9%)	5 (6.4%)	9 (10.2%)	19 (8.3%)
Death	6 (9.5%)	3 (3.8%)	4 (4.5%)	13 (5.7%)
Days of hospitalization ^#^	4.0 (2.0–13.9)	4.0 (2.0–9.3)	4.0 (2.0–21.0)	4.0 (2.0–18.2)
LDL > 100 ^##^	50/63 (79.4%)	60/78 (76.9%)	52/85 (61.2%)	162/226 (71.7%)
History of MI	2 (3.2%)	5 (6.4%)	18 (20.5%)	25 (10.9%)
History of Stroke ^##^	5/62 (8.0%)	3/77 (3.9%)	9/88 (10.2%)	17/227 (7.5%)

^#^ Data are reported as median (quantile 5%—quantile 95%); ^##^ Data are reported as a fraction (percent), where differences between the fraction denominator and the number of patients in the corresponding group indicate missing values in the data.

## Data Availability

The raw data supporting the conclusions of this article will be made available by the authors upon request.

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
