# Peer review of "The Elevated Inflammatory Status of Neutrophils Is Related to In-Hospital Complications in Patients with Acute Coronary Syndrome and Has Important Prognosis Value for Diabetic Patients"

_ijms, 2024, doi:10.3390/ijms25105107_

Round 1

Reviewer 1 Report

Comments and Suggestions for Authors

Barbu et al present an interesting work about the correlation between an inflammatory neutrophil phenotype and poor outcome in patients with myocardial infarction. They highlight that neutrophil activity had an important prognostic value for a complicated evolution in patients with pre-exisiting diabetes.  

The study contains many clinical data as well as in vitro data for the comparison of e.g. mRNA and protein levels of various cytokines and neutrophil activation marker. NET formation is also assessed using Sytox Green and a commercially available Neutrophil Elastase activity assay. Despite this broad approach, there are some comments I have that need to be addressed:

- Figure 4 and 5 have low quality

- Figure 6 C & D quantification says IL-1ß and Myd88 / tubulin but the corresponding Western Blot image says GAPDH

- Figure 6 and 7, does one dot represent one patient? Why do the corresponding Western blot original images not show different numbers of patients as the quantification? E.g. in A there are 14 dots for MI but in the original Western blot images there are only 8 MI patients shown

- Original Western Blot images should also show markers as it is difficult to assess the protein size otherwise

- Figure 7 B, please indicate the staining in the figure (e.g. blue Hoechst, green … as in Figure 8)

- Figure 7 C & D it seems that the labelling of the housekeeping gene in the y axis as C says Tubulin but shows GAPDH in the Western Blot image and vice versa for D

- The explanation for the patients group abbreviation for Figure 7F is missing in the legend

- Why was sometimes GAPDH and sometimes Tubulin used for the housekeeping gene and how comparable is that?

- Why does the GAPDH sometimes have a lot of bands (e.g. 6 B) and sometimes not? Was the antibody tested for specificity?

- Same for Tubulin, the Western Blot shows many sometimes many bands and sometimes only a single one (e.g. Figure 6A)

- For better understanding, the scale bar size for the fluorescent images in Figure 7 and 8 should be mentioned in the Figure legend as the number is very hard to read in the image itself

- Please list the primary and secondary antibodies used for Western Blot in the material and methods part

- L. 299-301 states that Figure 1 A shows significantly increased NETs levels in the serum of MI patients but the Figure legend to Figure 7 says for A “NETs levels in condition media of neutrophils from healthy 317 controls and from MI patients exposed for 1 h to 50mM PMA, as detected by Sytox green” (l. 317/318), so is it now serum or isolated neutrophils stimulated with PMA? Please clarify

Comments on the Quality of English Language

English Language is fine.

Author Response

                We appreciate constructive suggestions made by the reviewer. All the recommendations were well taken and we modified the manuscript accordingly.

All the changes are highlighted in grey in the revised form of the manuscript. 

Reviewer 1

Comments and Suggestions for Authors

Barbu et al present an interesting work about the correlation between an inflammatory neutrophil phenotype and poor outcome in patients with myocardial infarction. They highlight that neutrophil activity had an important prognostic value for a complicated evolution in patients with pre-exisiting diabetes.  

The study contains many clinical data as well as in vitro data for the comparison of e.g. mRNA and protein levels of various cytokines and neutrophil activation marker. NET formation is also assessed using Sytox Green and a commercially available Neutrophil Elastase activity assay. Despite this broad approach, there are some comments I have that need to be addressed:

- Figure 4 and 5 have low quality

Thank you for the observation. In the revised manuscript we provided high-quality versions of figures 4 and 5.

- Figure 6 C & D quantification says IL-1ß and Myd88 / tubulin but the corresponding Western Blot image says GAPDH.

Thank you for the observation. We are sorry for the errors that we made in the GraphPad program. In the revised manuscript we corrected the quantification graphs.

- Figure 6 and 7, does one dot represent one patient? Why do the corresponding Western blot original images not show different numbers of patients as the quantification? E.g. in A there are 14 dots for MI but in the original Western blot images there are only 8 MI patients shown

Yes, every dot represents one patient. The Western blot original images do not show results from all the patients as in the quantification graph,  because we did not include in the supplementary blot document, all the blots we performed.

- Original Western Blot images should also show markers as it is difficult to assess the protein size otherwise

            All the electrophoresis gels had molecular markers, but to minimize the blot area (to fit in our western blot boxes for antibody saving) we just cut out the markers lane. Before cutting the blots we made small bridges next to the marker strips (with a pen), to have exact indications about the molecular mass of markers and to further verify the molecular mass of the positive bands. For some smaller blots, the marker lane was not cut, and the negative images with the bridges of markers are shown in the blots document (Figure 6C).

- Figure 7 B, please indicate the staining in the figure (e.g. blue Hoechst, green … as in Figure 8)

As the reviewer suggested, we revised the legend of Figure 7B and included the information (grey highlighted) on page 12, line 338.

- Figure 7 C & D it seems that the labelling of the housekeeping gene in the y axis as C says Tubulin but shows GAPDH in the Western Blot image and vice versa for D

            We are sorry for these mistakes. In the revised manuscript we corrected them.  

- The explanation for the patients group abbreviation for Figure 7F is missing in the legend

Thank you for the observation. As the reviewer suggested we completed the legend of Figure 7 F  with the explanation regarding the patient groups (page 12, lines 343-344).

-  Why was sometimes GAPDH and sometimes Tubulin used for the housekeeping gene and how comparable is that?

For the majority of our blots, we have used GAPDH. The tubulin was used after we finished the GAPDH antibody and borrowed from our colleagues to finish the experiments. As the literature said, both GAPDH and tubulin are suitable and widely used as housekeeping genes, in most of the experimental conditions 10.1016/j.ab.2012.01.012, 10.1089/neu.2006.23.1794 .

- Why does the GAPDH sometimes have a lot of bands (e.g. 6 B) and sometimes not? Was the antibody tested for specificity?

Usually, we obtain a very nice, single-band for the GAPDH antibody we have used. Sometimes, depending on the blocking conditions, or if we did striping on a specific blot many times, these bands can appear. However, the antibody (ab9485, Abcam) which we and others have used intensively in the past, has high specificity. The antibody details are presented in the revised manuscript in the method section on page 19, line 583).

- Same for Tubulin, the Western Blot shows many sometimes many bands and sometimes only a single one (e.g. Figure 6A)

The tubulin antibody is also a good working antibody, displaying usually a single band. When multiple bands appear should be because of the increased number of striping we did on the same blot.

- For better understanding, the scale bar size for the fluorescent images in Figure 7 and 8 should be mentioned in the Figure legend as the number is very hard to read in the image itself

The reviewer's observation is pertinent. In the revised manuscript we mentioned the scale bar size in the legend of figures 7 and 8, on page 12, lines 338-339, and page 13, lines 363-364.

- Please list the primary and secondary antibodies used for Western Blot in the material and methods part.

As the reviewer suggested, in the method section, subsection 4.7. Western blot, we included the list of all the used primary and secondary antibodies. The paragraph below was added on pages 18-19, lines 579-586.

“The resulting blots were subsequently probed with specific primary antibodies directed against: IL-1β (MAB601, R&D Systems), S100A9 (PA1-46489, ThermoFisher), MCP-1 (ab9669, Abcam), MyD88 (AF2928, R&D Systems), p22phox (SC-271262, Santa-Cruz Biochtenology), MMP-9 (MAB936, R&D Systems), Neutrophil Elastase (MAB91671-100, R&D Systems), MPO (AF3667, R&D Systems), GAPDH (ab9485, Abcam) and β-tubulin (ab6046, Abcam), followed by secondary antibodies: anti-rabbit HRP-coupled (HAF008, R&D Systems), anti-mouse HRP-coupled (#31430, R&D Systems) or anti-goat HRP-coupled (HAF017, R&D Systems).”

 - L. 299-301 states that Figure 1 A shows significantly increased NETs levels in the serum of MI patients but the Figure legend to Figure 7 says for A “NETs levels in condition media of neutrophils from healthy 317 controls and from MI patients exposed for 1 h to 50mM PMA, as detected by Sytox green” (l. 317/318), so is it now serum or isolated neutrophils stimulated with PMA? Please clarify

Thank you for the observation. In Figure 7A the NETs levels were evaluated in the serum from control subjects or MI patients. We corrected the figure legend accordingly.

Reviewer 2 Report

Comments and Suggestions for Authors

Barbu and her colleagues performed a detailed population study in order to predict incidence of in-hospital cardiac complications among diabetic and pre-diabetic patients admitted to the hospital due to acute coronary syndrome.  The study includes reasonable numbers of patients, and analyses were done competently.  However, as listed below, there is a number of issues that need to be clarified, corrected, and/or discussed.

1.     This is a population study from which certain molecular and cellular differences were identified between diabetic and pre-diabetic groups.  The authors suggest that some of such differences may be used as biomarkers for predicting patients’ outcome, either favorable or unfavorable prognosis.  The differences are clearly detected when patients are analyzed as groups, but many figures seem to show that there is a huge overlap of individual data points (i.e. measured values of individual patients from each group) among the two groups of patients.  Is it not true, then, that the statistical differences found in the two groups may not be useful for predicting the outcome of a specific patient?  Can you assign some threshold values for predicting an outcome?  If not, the idea of biomarkers is weak because a biomarker should clearly distinguish individuals in one group from those in the other group.  This point should be discussed.  Again, the study was done competently, but I am not convinced about the results being used to discover new biomarkers for the intended prognosis.

2.     Abstract.  Please define all the abbreviations used here.

3.     Line19.  “Despite neutrophil involvement in inflammation and repair…”  Repairing what?  This phrase must be made clearer.

4.     Lines 111-112.  There are asterisks attached to p values.  This is done throughout the text.  These are unnecessary.  Please remove them., or explain what they indicate.

5.     Table 1.  LDL row.  Why do you have 50/63, 60/78, etc. instead of 50, 60, etc. like all others?  The same comment on History of Stroke, but here the denominators are different from LDL.  Why?  Please explain in the legend.  For Age and Days of Hospitalization, what do the numbers in parentheses indicate?  The range (i.e. the lowest to highest)?  If so, why the total column does not reflect this?  Please explain what these numbers are.

6.     Line 139-141.  “This section may be divided by subheadings. It should provide a concise and precise description of the experimental results, their interpretation, as well as the experimental conclusions that can be drawn.”  What is this?  It sounds like someone’s comment.

7.     Table 2.  The entire left-side column should not be in the bold font unless there is a good reason.

8.     Figure 3B and D. Explain what the red and blue dots are.

9.     Figure 7.  B.  Define the magnification bars (50 micrometer?).  C and D.  The loading control for C and D are GAPDH and tubulin, respectively, according to the gel, but the graphs show the reverse.  Please correct.

10. Figure 8.   Define the magnification bars.  Add magnification bars to the enlarged images in b1 and d1.  Explain in legend what arrows indicate.

11. Line 576.  “…anti-MPO human/mouse monoclonal antibody.”  This is not a monoclonal antibody but is a goat polyclonal antibody.  Correct the description.

12. Line 579.  “…chicken anti-mouse Alexa Fluor™…”  What was this reagent used for?  Explain.

13. Line627.  “The following supporting information can be…”  What do you refer to by “following”?   Describe what information is provided in the link.

Comments on the Quality of English Language

The journal editor should check for typos and grammar.

Author Response

We appreciate constructive suggestions made by the reviewer. All the recommendations were well taken and we modified the manuscript accordingly.

All the changes are highlighted in grey in the revised form of the manuscript. 

Comments and Suggestions for Authors

Barbu and her colleagues performed a detailed population study in order to predict incidence of in-hospital cardiac complications among diabetic and pre-diabetic patients admitted to the hospital due to acute coronary syndrome.  The study includes reasonable numbers of patients, and analyses were done competently.  However, as listed below, there is a number of issues that need to be clarified, corrected, and/or discussed.

  1. This is a population study from which certain molecular and cellular differences were identified between diabetic and pre-diabetic groups.  The authors suggest that some of such differences may be used as biomarkers for predicting patients’ outcome, either favorable or unfavorable prognosis.  The differences are clearly detected when patients are analyzed as groups, but many figures seem to show that there is a huge overlap of individual data points (i.e. measured values of individual patients from each group) among the two groups of patients.  Is it not true, then, that the statistical differences found in the two groups may not be useful for predicting the outcome of a specific patient?  Can you assign some threshold values for predicting an outcome?  If not, the idea of biomarkers is weak because a biomarker should clearly distinguish individuals in one group from those in the other group.  This point should be discussed.  Again, the study was done competently, but I am not convinced about the results being used to discover new biomarkers for the intended prognosis.

We are grateful for your insightful comments and for the opportunity to clarify the potential application of the molecular and cellular differences identified in our study as biomarkers for prognostic outcomes.

The reviewer's observation is correct in noting that while group-level differences between diabetic and pre-diabetic patients were evident, the individual data points do show substantial overlap. This is indeed an expected occurrence due to the inherent variability between patients and the multifactorial nature of diseases like diabetes.

We were indeed aware of the need for threshold values in the case of parameters intended to be used as individual biomarkers, and so we included Table 2 in our manuscript showing threshold values for each proposed biomarker, complete with their corresponding sensitivity and specificity. The thresholds were established based on Receiver Operating Characteristic (ROC) curve analysis, aiming to optimize the balance between sensitivity (true positive rate) and specificity (true negative rate). While no biomarker is perfect, the proposed thresholds are intended to provide the best possible differentiation between individuals with favorable and unfavorable prognoses based on our data. Moreover, not all molecules presented in the table are valid biomarkers, as shown by the low sensitivity and/or specificities of some/most of the molecules, and Table 2 should be regarded as an exploration of the biomarker utility of the molecules, rather than a list of biomarkers. Thus, we acknowledge that the presence of overlap does challenge the effectiveness of biomarkers when used in isolation, however, we propose that these molecules may be more powerful when used in combination, where the cumulative data can provide a better classification of prognosis, as further explored in the “Prognosis scores” section of our results.

Thus, as we do not intend to imply that all molecules in the table are potential biomarkers of patient prognosis, we changed the column names in Table 2 from “Biomarkers” to “Parameters”. We also replaced “Biomarkers” to “Parameters” in a couple of other places across the manuscript, keeping the term “biomarker” only for the best-performing parameters in predicting outcomes.

Also, to make our manuscript clearer, we moved Table 2 closer to the section “Prognosis scores” to which it refers and added the following paragraph in the results, on page 6, lines 184-192:

“Based on the sensitivity and specificity calculated at each parameter value, a Receiver Operating Characteristic (ROC) curve analysis was performed to identify threshold values as the values that best discriminate between favorable and unfavorable prognosis. Thus, for each parameter, the Youden j index (Sensitivity + Specificity -1) was calculated for each possible value that the parameter took, and the parameter value at the highest Youden j was used as the threshold. The results are shown in Table 2, along with the sensitivity and specificity associated with each threshold, offering insight into the true positive rate and the true negative rate achieved by the individual parameters.”

  1. Abstract.  Please define all the abbreviations used here.

Thank you for this observation. We introduced abbreviations in the abstract for left ventricular ejection fraction (LVEF), myocardial infarction (MI), Erythrocyte Sedimentation Rate (ESR), C-reactive protein (CRP), and reactive oxygen species (ROS).

  1. Line19.  “Despite neutrophil involvement in inflammation and repair…”  Repairing what?  This phrase must be made clearer.

            Thank you for this observation. We wanted to say that neutrophils are involved both in acute inflammation and in the resolution of inflammation. Moreover, the studies in the last years demonstrated an important role of neutrophils in reparation post-MI  (10.3390/cells10071676 ). To make the phrase clearer we changed it to “neutrophil involvement in inflammation and tissue repair.

Due to limitations imposed by the required abstract size, we can not describe here in further detail neutrophil functions such as phagocytosis, enzymes and cytokines release, Neutrophil Extracellular Traps (NETs) formation, etc., but we feel that the role of neutrophils in inflammation and tissue remodeling and repair is made clearer in the following introduction and discussion sections.

  1. Lines 111-112.  There are asterisks attached to p values.  This is done throughout the text.  These are unnecessary.  Please remove them., or explain what they indicate.

We removed asterisks attached to p values and only left the asterisks explaining the significance thresholds in figure legends.

  1. Table 1.  LDL row.  Why do you have 50/63, 60/78, etc. instead of 50, 60, etc. like all others?  The same comment on History of Stroke, but here the denominators are different from LDL.  Why?  Please explain in the legend.  For Age and Days of Hospitalization, what do the numbers in parentheses indicate?  The range (i.e. the lowest to highest)?  If so, why the total column does not reflect this?  Please explain what these numbers are.

The numbers like 50/63 and 60/78 represent fractions, where the numerator (e.g., 50, 60) is the number of patients who have the condition (like elevated LDL levels or a history of stroke), and the denominator (e.g., 63, 78) is the total number of patients for whom data were available. The reason for using this format rather than presenting only the numerator is to transparently reflect instances of missing data. For the other variables, where no missing data were recorded, the denominator is not shown, as that denominator is always the same as the number of patients in the corresponding group.

In Age and Days of Hospitalization rows, the numbers in parentheses following the median values represent the 5th and 95th percentiles, indicating the range within which the central 90% of our data falls. This provides a clearer view of the data distribution beyond the central tendency.

We have updated the Table 1 legend (page 7, lines 121-123) to include these explanations:

"#Data are reported as median (quantile 5% - quantile 95%).

##Data are reported as fraction (percent), where differences between the fraction denominator and the number of patients in the corresponding group indicate missing values in the data.”

  1. Line 139-141.  “This section may be divided by subheadings. It should provide a concise and precise description of the experimental results, their interpretation, as well as the experimental conclusions that can be drawn.”  What is this?  It sounds like someone’s comment.

We appreciate the reviewer's attention to detail in noting the presence of a section from the journal's template that was only meant for author guidance. This paragraph has now been removed from the document. Thank you for bringing this to our attention.

  1. Table 2.  The entire left-side column should not be in the bold font unless there is a good reason.

We revised Table 2, according to the reviewer's suggestion.

  1. Figure 3B and D. Explain what the red and blue dots are.

In figures 3B and D the red and blue dots represent patient data points, with red dots indicating patients with an unfavorable evolution and blue dots indicating patients with a favorable evolution. The background color of the decision space is also coded red for unfavorable and blue for favorable outcomes. This color coding helps to visually distinguish between the predictions and actual outcomes: red dots on a red background and blue dots on a blue background represent true positives and true negatives, indicating correct predictions of the patient's evolution. Conversely, red dots on a blue background are false negatives, and blue dots on a red background are false positives concerning predicting unfavorable evolution.

We added the following paragraph in the Figure 3 legend (page 8, lines 241-247):

 “The scatter plots in Fig. 3B and Fig. 3D show data from patients with unfavorable evolution as red dots and data from patients with favorable evolution as blue dots, while the decision space is shown as background color (red: unfavorable and blue: favorable evolution). Thus, red dots on a red background and blue dots on a blue background refer patients with correct evolution prediction (true positives and true negatives, respectively) red dots on a blue background show false negative results and blue dots on a red background show false positive results in predicting unfavorable evolution.”

We trust that this clarification enhances the understanding of our data visualization and the conclusions drawn from the machine learning analysis. We appreciate your attention to detail and your efforts to improve the clarity and accuracy of our work.

  1. Figure 7.  B.  Define the magnification bars (50 micrometer?).  C and D.  The loading control for C and D are GAPDH and tubulin, respectively, according to the gel, but the graphs show the reverse.  Please correct.

The reviewer's observation is pertinent. In the revised manuscript we mentioned the scale bar size in the legend of Figure 7, on page 12, lines 338-339, and corrected the errors regarding the GAPDH and tubulin.

  1. Figure 8.   Define the magnification bars. Add magnification bars to the enlarged images in b1 and d1.  Explain in legend what arrows indicate.

In the revised manuscript we added scale bars for the zoomed b1 and d1 images in Figure 8, added the scale size in the legend of Figure 8, on page 13, lines 363-364, and explained what the arrows indicate, page 13, lines 358-363.

  1. Line 576.  “…anti-MPO human/mouse monoclonal antibody.”  This is not a monoclonal antibody but is a goat polyclonal antibody.  Correct the description.

Thank you for this observation. We corrected the antibody description (page 19, line 608).

  1. Line 579.  “…chicken anti-mouse Alexa Fluor™…”  What was this reagent used for?  Explain.

The chicken anti-mouse Alexa Fluor is a secondary antibody coupled with Alexa Fluor fluorophore. This information was added in the revised manuscript on page 19, lines 610-611.

  1. Line627.  “The following supporting information can be…”  What do you refer to by “following”?   Describe what information is provided in the link.

This paragraph belongs to the IJMS journal template and we suppose that this will be the link provided by the journal for the supplementary material of the work.

Reviewer 3 Report

Comments and Suggestions for Authors

The manuscript “The elevated inflammatory status of neutrophils is related to in-hospital complications in patients with acute coronary syndrome and has important prognosis value for diabetic patients.” Describes the involvement of inflammatory parameters of MI patients with or without diabetes during the index event hospitalization and evaluates the pro-inflammatory phenotype of neutrophils isolated from MI patients with or without negative evolution.

The data presented herein are comprehensive, valuable, and engaging.

The entire manuscript is meticulously written, with a detailed and well-explained Materials and Method section and a thoroughly documented and explained Result section. The data presentation, both in the main text and supplementary data, is appropriately tabulated and illustrated. The results obtained in the study are adequately discussed, and the corresponding and up-to-date reference list is included in the manuscript.

However, there is a minuscule obstacle that must be corrected before the manuscript is ready for publication.

This includes:

The resolution of the variable names on the y-axis in Figures 4 and 5 is blurry and should be increased.

Acceptance of the manuscript in its present form (after correction of the Figure 4 and 5 y-axis resolution) is suggested.

Author Response

We thank the reviewer for his appreciation of our work and for the constructive suggestion he made. The recommendation was well taken and we modified the manuscript accordingly.

Comments and Suggestions for Authors

The manuscript “The elevated inflammatory status of neutrophils is related to in-hospital complications in patients with acute coronary syndrome and has important prognosis value for diabetic patients.” Describes the involvement of inflammatory parameters of MI patients with or without diabetes during the index event hospitalization and evaluates the pro-inflammatory phenotype of neutrophils isolated from MI patients with or without negative evolution.

The data presented herein are comprehensive, valuable, and engaging.

The entire manuscript is meticulously written, with a detailed and well-explained Materials and Method section and a thoroughly documented and explained Result section. The data presentation, both in the main text and supplementary data, is appropriately tabulated and illustrated. The results obtained in the study are adequately discussed, and the corresponding and up-to-date reference list is included in the manuscript.

However, there is a minuscule obstacle that must be corrected before the manuscript is ready for publication.

This includes:

The resolution of the variable names on the y-axis in Figures 4 and 5 is blurry and should be increased.

 Thank you for the observation. In the revised manuscript we provided high-quality versions of figures 4 and 5.

Acceptance of the manuscript in its present form (after correction of the Figure 4 and 5 y-axis resolution) is suggested.

Round 2

Reviewer 1 Report

Comments and Suggestions for Authors

Thank you for submitting the revised version of your manuscript. After discussion with the Editorial Team, we feel it is necessary to provide all original Western Blot images used for quantification in Figures 6 & 7 as supplemenatry file for a final decision on the manuscript. 

Author Response

We thank the reviewer for evaluating our manuscript and for constructive and helpful comments, which have improved the quality and clarity of our manuscript"

In accordance with the suggestions from both the reviewer and the Editorial team, we have included all the blots with samples used for statistical analysis in the supplementary materials and have revised the statistical graphs once more."

Reviewer 2 Report

Comments and Suggestions for Authors

Authors revised the paper nicely.  I have no more comments

Author Response

 We thank the reviewer for evaluating our manuscript and for constructive and helpful comments, which have improved the quality and clarity of our manuscript"